# MoViE: Mobile Diffusion for Video Editing

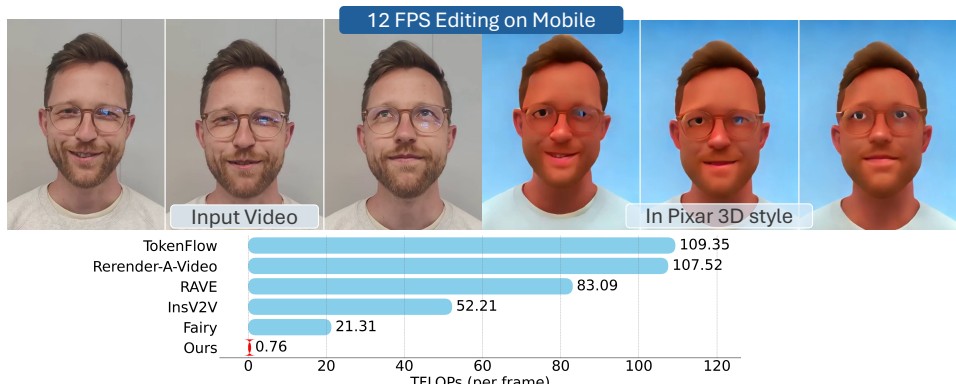

Figure 1: MoViE is a fast video editing model, capable of generating 12 frames per second on a mobile phone. It requires significantly fewer floating point operations (FLOPs) to edit a single video frame, making it considerably more efficient than competing methods while maintaining high editing quality.

## ABSTRACT

Recent progress in diffusion-based video editing techniques has shown remarkable potential and is being increasingly utilized in practical applications. However, these methods remain prohibitively expensive and particularly challenging to deploy on mobile devices. In this study, we introduce a series of optimizations that render mobile video editing feasible. Building upon the existing image editing model, we first optimize its architecture and incorporate a lightweight autoencoder. Subsequently, we propose *a new classifier-free guidance distillation* with multiple modalities, resulting in a $3\times$ on-device speed-up. Finally, we reduce the number of sampling steps to one ($10\times$ speed-up) by introducing *a novel adversarial distillation scheme* which preserves the *controllability* of the editing process in contrast to previous arts. Collectively, these optimizations enable video editing at an impressive 12 frames per second on mobile devices, while maintaining high editing quality.

## 1 INTRODUCTION

Diffusion-based generative models have shown impressive quality in image editing Hertz et al. (2022); Tumanyan et al. (2023); Brooks et al. (2022); Parmar et al. (2023). Building on these advancements, zero-shot video-based models extend these capabilities to the temporal dimension for video editing Wu et al. (2023a); Qi et al. (2023); Geyer et al. (2023); Ceylan et al. (2023); Wu et al. (2024); Liu et al. (2024). This is typically achieved by developing techniques to propagate information between frames ensuring that changes made in one frame are coherently reflected in subsequent frames. While these techniques achieve remarkable results in style transfer and attribute editing, they are computationally expensive, making them prohibitive to run on low-end devices.

While significant efforts have been made to reduce the computational cost of image diffusion models for on-device inference Li et al. (2024b); Zhao et al. (2023); Choi et al. (2023); Castells et al. (2024), there has been little effort toward developing video editing models on edge, mainly due to the high computational and memory costs. Inversion-based models Qi et al. (2023); Geyer et al. (2023); Kara et al. (2024) that extract source video features using DDIM inversion incur high computational and memory costs during inference. On the other hand, methods that propagate features from one or more frames to the other, a process known as cross-frame attention Wu et al. (2023a), also remain expensive to run on-device, as each frame must attend to multiple frames. For example, editing a 120-frame video at a resolution of $512 \times 384$ with Fairy Wu et al. (2024), a fast video-to-video (V2V)

model, takes 13.8 seconds on $8 \times$A100 GPUs. This is even higher for TokenFlow Geyer et al. (2023), an inversion-based V2V model, which takes 744 seconds Wu et al. (2024).

In this work, we introduce the first on-device video editing model capable of generating 12 frames per second on Xiaomi-14 Pro that uses Snapdragon 8 Gen. 3 Mobile Platform with a Qualcomm® Hexagon™ processor and Galaxy S25 Ultra with the Snapdragon 8 Gen. Elite Platform. To achieve this, we begin with a conventional video editing pipeline. Broadly speaking, diffusion editing pipelines Wu et al. (2023a; 2024) first encode source frames into the latent space using a Variational AutoEncoder (VAE) Pinheiro Cinelli et al. (2021). A denoising UNet model then iteratively denoises latents to generate edited latents, which are then decoded by the VAE decoder to produce the edited frames. In addition, many diffusion models use the classifier-free guidance technique Ho & Salimans (2022) to balance quality and diversity. Specifically, editing models utilize two guidance scales — image and text — which can be adjusted to trade off how strongly the generated image correspond with the source image and the edit instructions, respectively.

We identify factors that cause computational bottlenecks within an editing pipeline. First, we observed that the architectures of both the denoising UNet and the VAE encoder/decoder are major bottlenecks in the editing pipeline. Second, while classifier-free guidance technique enhances the model's controllability, it increases the computational cost three times as we need 3 forward passes per diffusion step to obtain the output. This is especially true for edge devices with limited memory that can handle only one sample at a time. Lastly, the intrinsic design of diffusion models necessitates iterative denoising to edit frames. Each denoising step involves complex computations, which makes the generation process expensive. To address the aforementioned challenges, we propose a series of contributions:

- We introduce the first on-device video-to-video diffusion model, capable of editing a 120-frame video at a resolution of $512 \times 384$ on a mobile device in 9.6 seconds, achieving a generation rate of 12 fps.

- We propose *a novel method to jointly distill multiple* guidance scales in the editing models, reducing the Number of Forward Evaluations (NFE) per diffusion step by 3 times without sacrificing the flexibility of classifier-free guidance.

- We present *a new adversarial distillation* recipe for reducing number of diffusion steps, ensuring the model's editing *controllability* is maintained post-distillation which is crucial for a video editing task and was not achieved before while reducing NFE of denoiser by 10 times.

- We factorize the optimization into four orthogonal components: architecture, autoencoder, guidance computation, and sampling, common across diffusion pipelines. This modular design enables targeted improvements and is readily transferable to other systems with similar building blocks.

Additionally, unlike some methods Wu et al. (2023a) that can generate only a limited number of frames due to memory constraints, our model can operate on long videos. This is because our model generate video frames independent of the video length, and its complexity increases linearly with the number of frames.

## 2 RELATED WORK

**Video generation.** Recent methods can be categorised into two groups. The first group enhances image models with temporal components, i.e. 3D convolutions, spatio-temporal transformers Ho et al. (2022a;b); Hong et al. (2022); Singer et al. (2022); Zhou et al. (2022), for modeling temporal dimension at the expense of the inflated computational footprint, along with the need for extensive training on large and diverse collections of data. In the effort to overcome these hurdles, zero-shot models Kahatapitiya et al. (2025), rely on training-free techniques for boosting temporal consistency Geyer et al. (2023); Khachatryan et al. (2023); Liu et al. (2024); Qi et al. (2023); Zhang et al. (2023). Common methods are, utilization of structural conditioning from a source video Zhang et al. (2023); Chen et al. (2023a), injection of diffusion inversion features into the generation process Geyer et al. (2023); Qi et al. (2023) and employing cross-frame attention using a set of key frames Qi et al. (2023); Zhang et al. (2023); Wu et al. (2023a); Esser et al. (2023). Our method aligns with this direction but is computationally much lighter, making it suitable for low-end edge devices.

**Optimized diffusion.** Optimizing diffusion mainly entails the reduction of required steps for traversing the probability path from pure noise to the denoised sample, as well as reducing the cost of each step. In line with the former are the utilization of higher order solvers Zhang & Chen (2022); Lu et al. (2022a;b), the straightening of underlying ODE trajectories or direct mapping to data via Rectified Flows Liu et al. (2022); Liu (2022); Zhu et al. (2025) and consistency models Lu & Song (2024); Song et al. (2023); Song & Dhariwal (2023), respectively and the employment of progressive-step distillation Li et al. (2024b); Salimans & Ho (2022b); Meng et al. (2023) or adversarial training Wang et al. (2022); Sauer et al. (2024; 2025); Zhang et al. (2024b) for fewer or even single step diffusion. On the other hand, quantization He et al. (2024); Pandey et al. (2023); Shang et al. (2023) and pruning Li et al. (2024b); Choi et al. (2023) have been used, as well as extensive research conducted on simplifying the denoiser itself Dockhorn et al. (2023); Kim et al. (2023); Habibian et al. (2024). Our work incorporates both architectural optimization and a reduction in the number of steps. While extending Li et al. (2024b) with multimodal distillation, we address a limitation of Sauer et al. (2024); Zhang et al. (2024b), by restoring control over text and image guidance, the level of adherance to text and image conditioning, respectively.

**On-device generation.** The abundance of edge devices, along with reduced cost and more privacy-secure inference on edge comparing to cloud-based approaches, have led to recent works targeting mobile devices and their NPUs Castells et al. (2024); Chen et al. (2023b); Choi et al. (2023); Li et al. (2024b); Zhao et al. (2023). While there has been progress in the video domain with fast zero-shot video editing models Zhang et al. (2024a); Kara et al. (2024); Wu et al. (2024), there have been no attempts to implement video editing models on device due to their high computational and memory requirements. Our method pushes the boundaries by unlocking on-device zero-shot video editing.

## 3   MOVIE

Our goal is to develop an on-device video-editing model that optimizes the trade-off between efficiency and quality. To achieve this, we follow four stages: (1) selecting a base model, (2) introducing a mobile-friendly denoiser called Mobile-Pix2Pix, (3) reducing the cost of classifier-free guidance through multimodal guidance distillation, and (4) reducing the number of diffusion sampling steps using adversarial step distillation.

### 3.1   BASE MODEL

Zero-shot video editing using diffusion models have achieved impressive results. The common approach among such models is to employ an off-the-shelf image editing model and replace self-attention layers with cross-frame attention to ensure temporally consistent and coherent frame generation. In specific, each self-attention layer receives the latent from previous layer and linearly projects it into query, key and value $Q, K, V$ to produce the output by Self-Attn$(Q, K, V) = Softmax(QK^T/\sqrt{d})V$. Cross-frame attention extends the idea by concatenating keys and values from one or more other frames, commonly called as anchors, resulting in CrossFrame-Attn$(Q, K', V')$ $= Softmax(Q[K_1; , ...; K_t]^T/\sqrt{d})[V_1; ...; V_t]$. For example, Rerender-A-Video Yang et al. (2023) uses Stable Diffusion Rombach et al. (2022) as image editing model and uses first and previous frames as anchors to attend to the current frame. Fairy Wu et al. (2024) uses InstructPix2Pix Brooks et al. (2022) as an instruction-based image editing model and uniformly selects 3 frames with equal intervals as anchors. Following the same direction, we use InstructPix2Pix as the base image model. InstructPix2Pix processes a source image and a text instruction for the desired edit. The text is encoded using a text encoder (like CLIP Radford et al. (2021)), and the source image is converted into latent representations via a Variational Auto-Encoder (VAE). A conditional U-Net model then refines these latents iteratively for a fixed number of sampling steps to produce the edited image, which is finally decoded back to pixel space using VAE decoder. Frames are generated in 10 diffusion steps, with the middle frame used as the sole anchor. In the subsequent sections, we detail our optimization strategies for the base model.

### 3.2   MOBILE-PIX2PIX

The InstructPix2Pix pipeline is computationally intensive. For instance, denoising one step of a batch containing a single sample with UNet requires approximately 600 GFLOPs. For 10 steps and three forward passes needed for classifier-free guidance, this totals around 18.05 TFLOPs. Additionally, encoding and decoding an image of resolution $480 \times 480$ using VAE incurs cost of 3.2 TFLOPs.

In our effort to optimize InstructPix2Pix pipeline for efficient video editing on mobile devices, we implemented two changes to its pipeline. First, similar to Li et al. (2024b), we remove expensive self-attention and cross-attention layers at the highest resolutions, in both encoder and decoder part of the UNet. Removing them leads to 12% reduction in FLOPs.

Another bottleneck in the editing pipeline is the encoding and decoding of latents using VAE. To mitigate this, we utilize the Tiny Autoencoder for Stable Diffusion (TAESD) Bohan, a deterministic and lightweight autoencoder composed of a series of residual blocks. Its encoder is a distilled version of the VAE encoder, whereas the decoder has been trained as a stand-alone GAN, optimized with adversarial and reconstruction losses. TAESD's reduced memory and computational footprint renders it as a perfect candidate for limited-budget applications (i.e. on-edge). Moreover, its shared latent space with VAE enables hybrid autoencoding, with VAE decoding TAESD's latents and vice versa. Incorporating TAESD in our pipeline leads to a 92.6% reduction in FLOPs compared to original InstructPix2Pix VAE with a modest drop in quality (see Sec. 4.3).

---

**Algorithm 1** Multimodal Guidance Distillation

---

**Require:** Teacher $\epsilon$, Student $M_\theta$, dataset $D$, loss weighting term $\lambda$, learning rate $\gamma$
1: **while** not converged **do**
2:     $c_I, c_T, x \sim D$ ▷ Sample input image, prompt, edited image
3:     $t \sim U[0, 1000]$           ▷ Sample time
4:     $s_I \sim U[1, 3]$    ▷ Sample image guidance scale
5:     $s_T \sim U[2, 14]$    ▷ Sample text guidance scale
6:     $\epsilon_n \sim N(0, I)$         ▷ Sample noise
7:     $x_t = \alpha_t x + \sigma_t \epsilon_n$    ▷ Add noise to data
8:     Obtain $\tilde{\epsilon}$ from Eq. 1         ▷ CFG
9:     $L = \lambda \|M_\theta(x_t, c_I, c_T, s_I, s_T) - \tilde{\epsilon}\|_2^2$  ▷ Loss
10:    $\theta \leftarrow \theta - \gamma \nabla_\theta L$        ▷ Optimization
11: **end while**

---

### 3.3 Multimodal Guidance Distillation

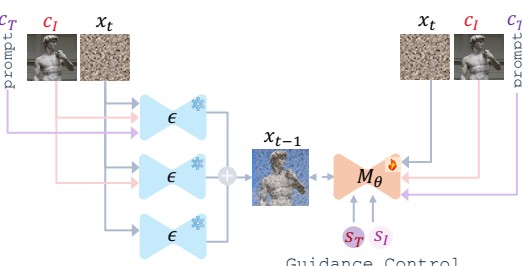

Figure 2: Multimodal Guidance Distillation Overview: Standard classifier-free guidance inference pipeline (left) with two input conditionings (image and text) requires three inference runs on each diffusion step. Our distilled pipeline (right) incorporates guidance scales $s_I$ and $s_T$ into UNet and only performs one inference run.

Classifier-free guidance (CFG) Ho & Salimans (2022) is a popular technique to improve the quality of text-to-image generative models. This is achieved by a linear combination of estimates from text-conditional and unconditional models weighted by a so-called guidance scale. Several works extend and utilize CFG with multiple conditioning modalities for image-to-image editing Brooks et al. (2022), editing 3D scenes with text Haque et al. (2023), novel view synthesis Liu et al. (2023), and video generation Kondratyuk et al. (2023). InstructPix2Pix Brooks et al. (2022) also utilizes CFG with multiple modalities and performs three separate forward passes with varied combinations of image condition $c_I$ and text condition $c_T$ to generate the final output:

$$\tilde{\epsilon}(x_t, c_I, c_T) = \epsilon(x_t, \varnothing, \varnothing) + s_I \left(\epsilon(x_t, c_I, \varnothing) - \epsilon(x_t, \varnothing, \varnothing)\right) + s_T \left(\epsilon(x_t, c_I, c_T) - \epsilon(x_t, c_I, \varnothing)\right) \quad (1)$$

where $\epsilon(.)$ is a diffusion model and $s_I$ and $s_T$ are image and text guidance scales respectively and can be adjusted to trade off the edited image fidelity to the source image or the edit instruction.

While CFG allows for a trade-off between fidelity and diversity, it is computationally expensive due to the multiple forward passes needed to generate the final output (see Eq. 1). The compute costs can quickly become a bottleneck in applications that require multiple conditioning modalities. To reduce such costs, Meng et al. (2023) distills CFG into a model that requires only one forward pass per diffusion step. However, their work presents a distillation strategy for only a single conditioning modality, that is, text.

In this work, we extend the concept of CFG distillation to *multiple* modalities, specifically text and image conditionings, to further reduce the cost of InstructPix2Pix pipeline. As shown in Fig. 2, we distill $\epsilon(.)$ in Eq. 1 to a single forward pass using the loss function:

$$\mathbb{E}_{c_I, c_T \sim p_{\text{data}}} \left[ \|M_\theta(x_t, c_I, c_T, s_I, s_T) - \tilde{\epsilon}(x_t, c_I, c_T)\|_2^2 \right],$$

where $M_\theta(.)$, a student model, receives guidance scales $s_I$ and $s_T$ along with the conditions. This is done first by projecting these scales to embeddings using sinusoidal timestep embeddings Ho et al. (2020). Each ResNet block is then modified to accept these embeddings similarly to timestep embedding, which are further processed through a linear layer and nonlinearity before being added to the latent representation. Note that both image and text guidance scales are distilled at once through our procedure, avoiding the time and compute costs of sequential distillation. We add details for the method in Alg. 1. As a result, the distilled model $M_\theta(.)$ allows for a single forward pass in each diffusion step ($3\times$ speed-up) without compromising the editing quality or control, i.e. it still supports any guidance scale values $s_I$ and $s_T$ after distillation (See Fig. 5).

## 3.4 ADVERSARIAL STEP DISTILLATION

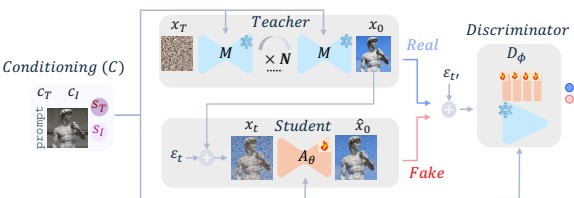

Figure 3: Adversarial Distillation: We distill a multi-step teacher into a single step student using adversarial losses. Unlike existing adversarial distillation approaches Sauer et al. (2024); Zhang et al. (2024b) that forego guidance flexibility for faster sampling, we preserve guidance strength property during adversarial training by providing the synthetic latent $x_t$ from teacher's denoising process and conditioning the student on the corresponding guidance scales.

Having reduced the computational footprint of each step, in this section, we seek to distill the multi-step model obtained in Sec. 3.3 into a single-step model to establish an efficient on-device editing pipeline. Improving sampling efficiency of diffusion models has been the focus of several works Li et al. (2024b); Salimans & Ho (2022b); Lu & Song (2024). Recently, adversarial distillation diffusion models that require a single denoising step have shown impressive results for text-to-image (T2I) and image-to-image (I2I) editing applications Sauer et al. (2024; 2025); Zhang et al. (2024b). However, the classifier-free guidance property is usually ignored during training Sauer et al. (2024); Zhang et al. (2024b). For example,

LADD Sauer et al. (2024) foregoes the flexibility of text and image guidance. This oversight of guidance scales leads to models that *do not have any control* over edit strength during inference, which is imperative in video editing. In this work, we propose *a novel adversarial training algorithm* that distills a multi-step teacher into a single-step student while preserving its *editing controllability* with respect to the input modalities. The key insight is to use the latents generated from the teacher's denoising process as an input to the student as well as discriminator and condition the student on the corresponding guidance scales used by the teacher.

Our adversarial distillation setup is outlined in Fig. 3. Student $A_\theta$ is initialized from a pre-trained teacher $M$ of the previous stage. Discriminator $D_\phi$ consists of a frozen feature extractor, initialized from the encoder arm of the teacher and a set of trainable spatial heads applied to each layer of feature extractor. Detailed architecture of spatial heads can be found in the supplementary.

During adversarial training, the *real* edited sample $x_0$ is obtained from the multi-step teacher. The student receives a noisy sample $x_t$ generated by the forward diffusion process to $x_0$ using student noise schedule. To retain controllability, the corresponding guidance scales $s_I, s_T$ are also provided to the student. The student then generates the *fake* samples $\hat{x}_0$ which is then evaluated by the discriminator. Similar to Sauer et al. (2024), we first add noise to $x_0$ and $\hat{x}_0$ before feeding them to the discriminator. For more details see Alg. 2. The discriminator distinguishes between *real* and *fake* samples, providing feedback to the student model. This process enhances the student

---

**Algorithm 2** Adversarial Distillation

**Require:** Teacher model $M$, Student model $A_\theta$, Discriminator $D_\phi$, dataset $D$, loss weighting terms $\lambda_{loss}$, learning rate $\gamma$ and $\gamma_d$ for $A$ and $D_\phi$, preconditioning functions $c_{in}$ and $c'_{in}$ Karras et al. (2022a), $R_1$ is the gradient penalty Mescheder et al. (2018)

1: **while** not converged **do**
2:     $c_I, c_T \sim D, s_I \sim U[1,3], s_T \sim U[2,14]$
3:     $C = [c_I, c_T, s_I, s_T]$
4:     $\epsilon \sim N(0,I); x_0 = M(\epsilon, C)$   ▷ Real sample from teacher's denoising process
5:     $t \sim U[0,8], \epsilon \sim N(0,I)$
6:     $x_t = c_{in}(t) * (\alpha_t x_0 + \sigma_t \epsilon)$   ▷ Add noise to real
7:     $\hat{x}_0 = A_\theta(x_t, C, t)$   ▷ Fake (Student) sample
8:     $t' \sim U[0,1000], \epsilon' \sim N(0,I)$
9:     $x_{t'} = c'_{in} * (\alpha_{t'} x_0 + \sigma_{t'} \epsilon')$   ▷ Add noise to real
10:    $\hat{x}_{t'} = c'_{in} * (\alpha_{t'} \hat{x}_0 + \sigma_{t'} \epsilon')$   ▷ Add noise to fake
11:    $L = \mathbb{E}_{t',x_0}[max(0, 1 + D_\phi(x_{t'}, C)) + \lambda_{r1} R_1] + \mathbb{E}_{t,t',x_0}[max(0, 1 - D_\phi(\hat{x}_{t'}, C))]$
12:    $\phi \leftarrow \phi - \gamma_d \nabla_\phi L$   ▷ Update Disc.
13:    Repeat Steps 8 and 10
14:    $L = \lambda_{mse} \|\hat{x}_0 - x_0\|_2^2 + \lambda_{gen} \mathbb{E}_{t,t',x_0}[D_\phi(\hat{x}'_t, C)]$
15:    $\theta \leftarrow \theta - \gamma \nabla_\theta L$   ▷ Update Student
16: **end while**

model's ability to produce high-quality, controlled edits in just one sampling step ($10\times$ speed-up) with a greater flexibility compared to previous methods.

# 4 EXPERIMENTS

## 4.1 EXPERIMENTAL SETUP

**Datasets and metrics.** In our work, we utilize the InstructPix2Pix dataset Brooks et al. (2022) for finetuning and distillation. The dataset has around $300k$ samples, each sample consisting of (source image, edit instruction, and edited image) triplet. For evaluation and ablations, we use the validation set consisting of around $5k$ samples. Following Brooks et al. (2022), we utilize CLIP-Image similarity and directional CLIP similarity to estimate fidelity of edited image to the input image and edit prompt respectively. Higher values for these metrics indicate superior performance. Following Cheng et al. (2023); Singer et al. (2024), for comparison with the state-of-the-art, we employ the Text-Guided Video Editing (TGVE) benchmark Wu et al. (2023b). The dataset comprises of 76 videos where each video includes 4 prompts on editing style, background, object, and multiple attributes. We measure PickScore Kirstain et al. (2023) and CLIPFrame Hessel et al. (2022) to evaluate the quality and consistency of the edits respectively. To measure number of floating point operations (FLOPs), we use DeepSpeed (Rasley et al., 2020, v0.15.2). Latencies are measured on a single NVIDIA® A100 GPU, on a Xiaomi-14 Pro encompassing a Snapdragon 8 Gen. 3 Mobile Platform, and Galaxy S25 Ultra with the Snapdragon 8 Gen. Elite Platform, for GPU and Phones respectively. Reported latencies correspond to a single-frame denoising.

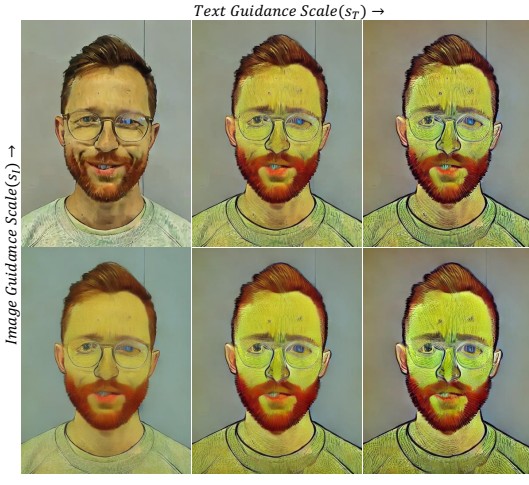

*Text Guidance Scale($s_T$) →*

*Image Guidance Scale($s_I$) →*

Figure 4: MoViE at text guidance $[4.0, 8.0, 12.0]$ and image guidance $[1.25, 1.75]$. Our adversarial training maintains guidance scales, allowing us to control edit strength during inference.

**Implementation details.** To obtain an efficient and mobile-friendly model of InstructPix2Pix, we make sequential architectural and inference optimizations as follows. **Stage 1**: We begin with the text-to-image Stable Diffusion (SD) UNet V-1.5 UNet. Inspired by Li et al. (2024b), we remove the costly self-attention and cross-attention layers at the highest resolution and finetune the model. **Stage 2**: Similar to Brooks et al. (2022), we add image conditioning layers and transform the T2I model to an I2I editing model and train on InstructPix2Pix dataset for 20k iterations, resulting in Mobile-Pix2Pix. **Stage 3**: We distill our multimodal guidance scales as described in Alg. 1. We use 5k iteration for training where the teacher is a model from the previous stage. **Stage 4**: For guidance-preserving adversarial distillation, we first convert the model obtained from the previous stage to $v - prediction$ as we find that it helps to reduce noisy artifacts during adversarial training. We then distill a multi-step teacher model into a single-step student using a weighted combination of $MSE$ and $GAN$ loss for 20k iterations. The detailed training steps are shown in Alg. 2, and further details can be found in the supplementary.

## 4.2 RESULTS

**MoViE.** We compare the impact of the previously introduced optimization stages on editing quality in terms of directional CLIP similarity and CLIP-Image similarity. As shown in Fig. 5, our first optimization stage, Mobile-Pix2Pix, achieves comparable quality as the original model while being $24.4\%$ cheaper in terms of FLOPS and $73\%$ faster on-device (see Tab. 1). With multimodal guidance distillation, a single forward pass is now required, thus our pipeline becomes $66\%$ cheaper computation-wise compared to previous stage, still at the cost of negligible quality drop. Finally, our adversarial distillation technique further improves UNet latency by a factor of 10 and end-to-end latency by a factor of 7.5, enabling on-device editing at 12 frames per second. In the last stage, we observe a slight decline in quantitative metrics; however, the substantial gains in efficiency justify the

| Method | Steps | PickScore↑ | CLIPFrame↑ | TFLOPs (per frame) | Latency (GPU) | Latency (Phone) |
|---|---|---|---|---|---|---|
| Fairy Wu et al. (2024) | 10 | 19.80 | 0.933 | - | - | - |
| TokenFlow Geyer et al. (2023) | 50 | 20.49 | 0.940 | 109.35 | 2.45s | - |
| Rerender-A-Video Yang et al. (2023) | 20 | 19.58 | 0.909 | 107.52 | 2.13s | - |
| ControlVideo Zhang et al. (2023) | 50 | 20.06 | 0.930 | 89.49 | 5.63s | - |
| InsV2V Cheng et al. (2023) | 20 | 20.76 | 0.911 | 52.21 | 2.70s | - |
| RAVE Kara et al. (2024) | 50 | 20.35 | 0.932 | 83.09 | 4.31s | - |
| FLATTEN Cong et al. (2023) | 50 | 20.47 | 0.925 | 205.90 | 15.96s | |
| VidToMe Li et al. (2024a) | 50 | 20.33 | 0.923 | 198.12 | 5.43s | |
| EVE Singer et al. (2024) | - | 20.76 | 0.922 | - | - | - |
| Base Model | 10 | 20.34 | 0.943 | 21.31 | 1.37s | 7s |
|   + Mobile-Pix2Pix | 10 | 19.43 | 0.922 | 16.10 | 1.06s | 1.9s |
|    + Multi-Guidance Dist. | 10 | 19.60 | 0.919 | 5.50 | 0.82s | 0.6s |
|     + Adversarial Distillation (MoViE) | 1 | 19.40 | 0.913 | 0.76 | 0.11s | 0.08s |

Table 1: End-to-end FLOPs and latency of video editing models on $480 \times 480$ resolution on **TGVE** benchmark, normalized per frame. On-device latencies correspond to $512 \times 384$ frames. PickScore and CLIPFrame for competing methods (except RAVE) are taken from the InsV2V Cheng et al. (2023).

trade-off, as evidenced by Tab. 1 and the qualitative results in Fig. 8. Finally, in Fig. 4 we show that our adversarial distillation technique retains the flexible editing property allowing us to control the strength of edit at inference.

**SOTA comparison.** In Tab. 1 we compare several state-of-the-art methods with ours in terms of quality and computation cost. These methods include: Fairy Wu et al. (2024), an anchor-based cross-frame attention method for fast and coherent video-to-video synthesis; TokenFlow Geyer et al. (2023), a video editing method that propagates inversion features across frames; Rerender-A-Video Yang et al. (2023), a zero-shot text-guided model with temporal-aware patch matching; ControlVideo Zhang et al. (2023), a training-free controllable model with cross-frame interaction; InsV2V Cheng

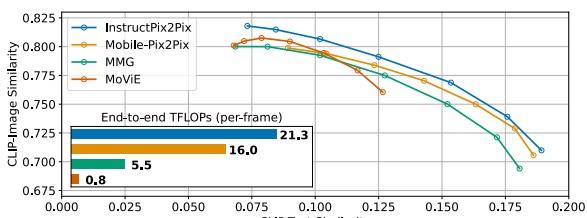

Figure 5: CLIP metrics for InstructPix2Pix, Mobile-Pix2pix, Multi-modal Guidance (MMG) Mobile-Pix2pix and MoViE. As shown in the graphs, proposed optimizations improve the efficiency greatly with minimum quality drop.

et al. (2023), a consistent video editing model, EVE Singer et al. (2024), a frame editing method with video adapters; RAVE Kara et al. (2024) that uses a novel noise shuffling strategy for fast zero-shot video editing; FLATTEN Cong et al. (2023), a training-free method utilizing optical flow-guided attention to maintain visual consistency across frames; and VidToMe Li et al. (2024a) that enhances temporal consistency by merging self-attention tokens across frames. We found these methods to be highly computationally intensive in terms of FLOPs, making them unsuitable for mobile video editing applications. In contrast, our series of optimizations shows significant improvements in both TFLOPs and GPU latency per frame while maintaining good quality edits, as evidenced by PickScore Kirstain et al. (2023) and CLIPFrame Hessel et al. (2022) metrics in the TGVE benchmark Wu et al. (2023b). Unlike LADD Sauer et al. (2024), our adversarial distillation retains editing controls, allowing to adjust edit strength during inference per sample, which improves PickScore and CLIPFrame even further from 19.40 to 20.02 and from 0.913 to 0.943 respectively. Finally, we profile the latency of our method on a Xiaomi 14 Pro with Snapdragon® 8 Gen. 3 Mobile Platform as well as Galaxy S25 Ultra with the Snapdragon® 8 Gen. Elite Platform. As shown in Tab. 1, our optimizations achieve an impressive on-device denoising frame rate of 12.

**Qualitative results.** First, we present the qualitative results of MoViE on DAVIS Perazzi et al. (2016) videos, as illustrated in Fig. 6. Our method effectively performs both global edits and fine-grained attribute editing with high efficiency. In Fig. 8, we compare our model to the base model, demonstrating enhanced efficiency while maintaining editing quality. Finally, in Fig. 9, we compare MoViE with several state-of-the-art methods on two DAVIS Perazzi et al. (2016) videos featuring challenging editing prompts. To ensure a fair comparison, each test prompt was adapted to the input style (e.g., instructional or descriptive) that best suited each method. In Fig. 9, prompts are shown for illustration; the actual input to each model may differ, as we used the formulation that yielded the best performance for each case. In the first scenario, which involves changing the hair color of a woman, InsV2V Cheng et al. (2023) and TokenFlow Geyer et al. (2023) performed relatively well, whereas Rerender-A-Video Yang et al. (2023) introduced numerous unnecessary changes. In the

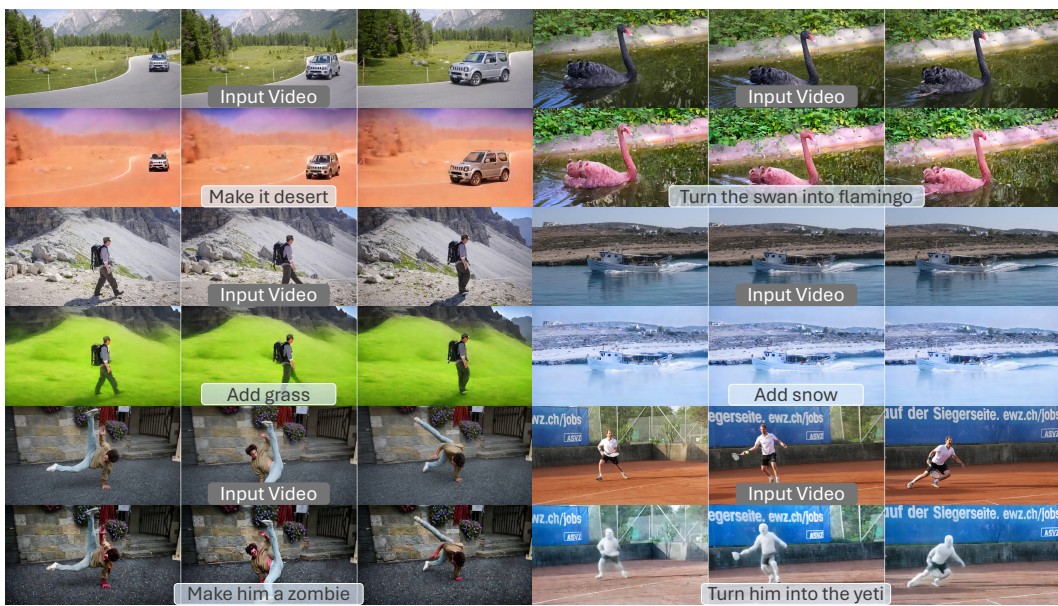

Figure 6: Qualitative results of MoViE on DAVIS. Our method can handle complex global edits as well as perform more nuanced attribute editing while requiring very few computational resources. Please refer to the supplementary for video results.

| Type of Ablation | Ablation detail | CLIP-Image |
|---|---|---|
| Noise Distribution | $m = 0, s = 1$ | 0.759 |
| | $m = -1, s = 1$ | 0.769 |
| | $m = -1, s = 2$ | 0.765 |
| Head Architecture | No guidance conditioning | 0.781 |
| | With guidance conditioning | 0.786 |

Table 2: Ablation study on discriminator design choices during adversarial training.

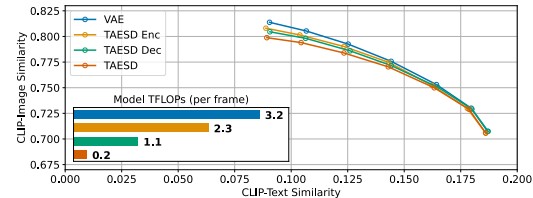

Figure 7: CLIP metrics for various autoencoder configurations. A substational FLOPs reduction can be achieved by incorporating TAESD, with minimal drop in editing performance.

second scenario, where a fire effect was requested to be added in, most methods failed except for InsV2V Cheng et al. (2023). For both prompts, our method produced good quality edits, indicating its competitiveness with more expensive video editing algorithms. Please refer to the supplementary for video results.

## 4.3 ABLATIONS

**Adversarial Training.** To confirm the effectiveness of our design choices, we conduct an ablation study for both the discriminator and generator setup. We use the guidance-distilled Mobile-Pix2Pix model from **Stage 3** that has been finetuned for $v - prediction$. We evaluate the models for a single sampling step using the InstructPix2Pix validation dataset and report CLIP-Image similarity metric in Tab. 2 for guidance scale of 7.5 and image guidance scale of 1.2. For the discriminator noise (see Alg. 2), we find similar observations as Sauer et al. (2024); Zhang et al. (2024b), where the noise distribution affects the edit quality. As shown, setting the mean and standard deviation of noise distribution to $-1$ and $1$, respectively gives the best results. We also observe that using guidance conditioning on discriminator heads provides better prompt adherence and flexibility although the CLIP-Image scores do not appear to vary significantly.

**Impact of TAESD.** Incorporating TAESD significantly reduces FLOPs, as shown in Tab. 1 and Fig. 7, with a moderate drop in CLIP scores. The shared embedding space allows combining VAE's and TAESD's encoders-decoders for partial gains in the diffusion pipeline. As seen in Fig. 7, allocating more budget to the decoder, although more expensive, provides better reconstruction. We use TAESD for both encoding and decoding to achieve a good balance between quality and efficiency.

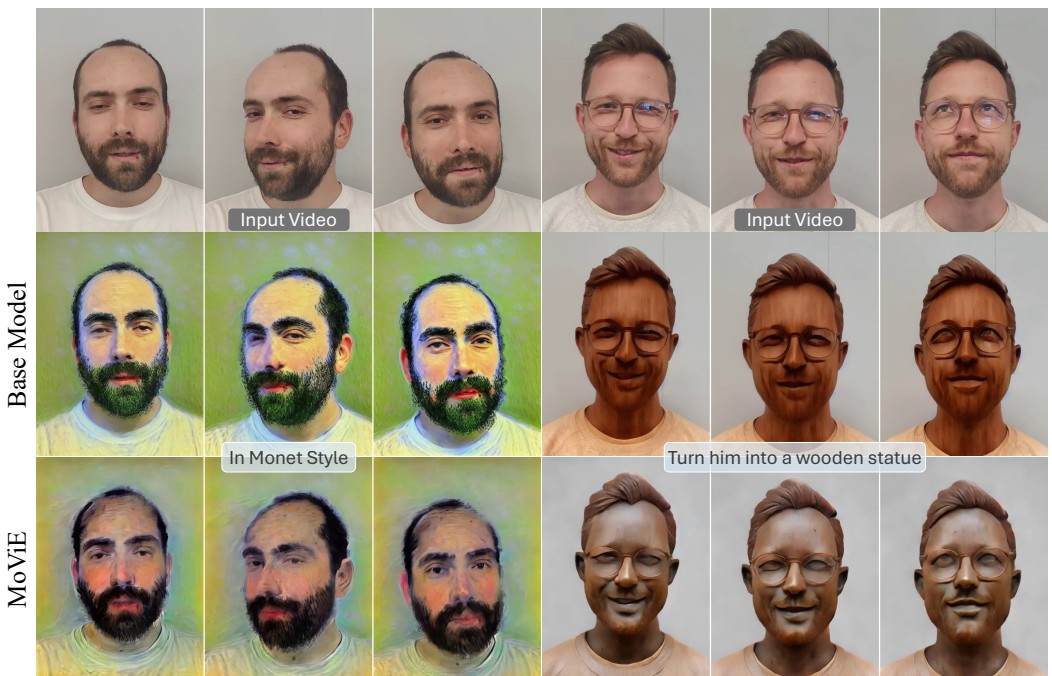

Figure 8: Qualitative comparison of our method to the base model. The efficiency is greatly improved whereas quality is not compromised both for style transfer and attribute edits. Please refer to the supplementary for video results.

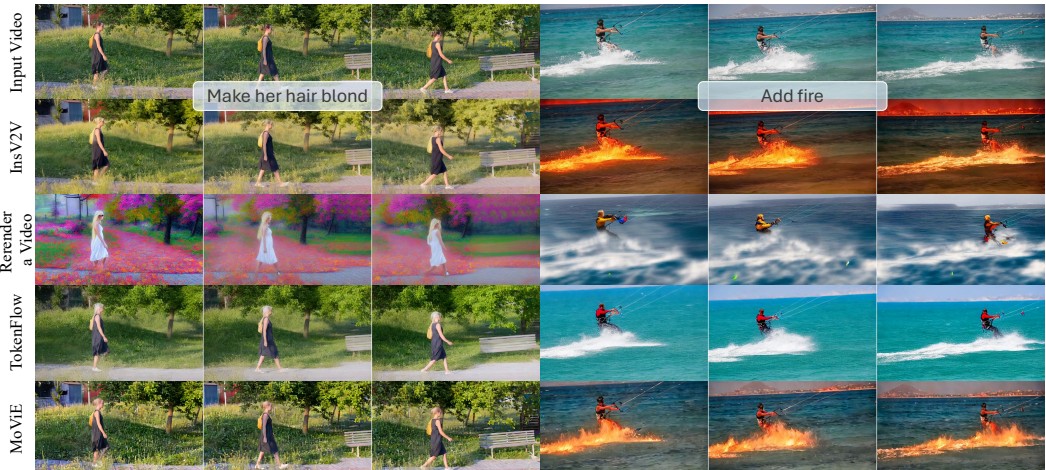

Figure 9: Qualitative comparison of our method MoViE to other SOTA video editing algorithms. We evaluate on two challenging editing scenarios. MoViE produces good quality edits yet far outperforms other competing methods in terms of efficiency. Please refer to the supplementary for video results.

## 5 CONCLUSION

In this work, we have explored several optimizations to accelerate diffusion-based video editing. Firstly, we introduced an architectural enhancement of the denoising UNet and a lightweight autoencoder, resulting in our Mobile-Pix2Pix model. Secondly, we performed a novel multimodal guidance distillation which consolidates classifier-free guidance inference with text and image as input conditionings into one forward pass per diffusion step, achieving a $3\times$ increase in inference speed. Lastly, we optimized the number of diffusion steps to one through a new adversarial distillation procedure ($10\times$ speed-up), while maintaining the controllability of edits unlike earlier approaches. These optimizations enable 12 frames per second video editing on-device, marking a significant milestone towards real-time text-guided video editing on mobile platforms.

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

## A SUPPLEMENTARY MATERIALS

### A.1 WEBPAGE AND HUMAN EVALUATION

**Webpage.** Please refer to the attached standalone html file (by opening the website/index.html) to visually examine the generated videos. Best viewed using Chrome, FireFox, or Microsoft Edge. The videos displayed on the html page are available at the website/static/videos folder for closer inspection.

**Human Evaluation.** We conducted an A/B comparison between MoViE and three state-of-the-art (SOTA) methods: TokenFlow Geyer et al. (2023), InsV2V Cheng et al. (2023), and Rerender-A-Video Yang et al. (2023). Participants were asked to compare the overall quality of the generated videos, choosing between: 1) video 1 is better, 2) video 2 is better, or 3) both are equal. Each participant compared 24 video pairs, with each pair evaluated by 32 participants. As shown in Fig. 10, participants generally preferred our method over the SOTA methods. Our model significantly outperformed Rerender-A-Video and was reasonably preferred over TokenFlow and InsV2V. However, due to the limited sample size, we cannot draw definitive conclusions.

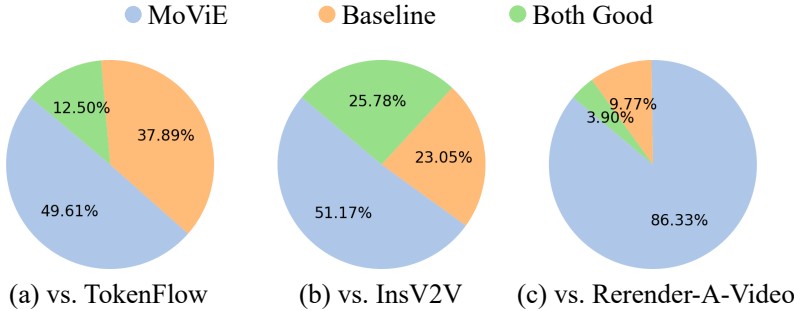

(a) vs. TokenFlow      (b) vs. InsV2V      (c) vs. Rerender-A-Video

Figure 10: Human Evaluation results comparing MoViE to TokenFlow Geyer et al. (2023), InsV2V Cheng et al. (2023), and Rerender-A-Video Yang et al. (2023).

### A.2 TRAINING DETAILS

**Spatial Head Architecture.** As discussed in Section 3.4, discriminator $D_\phi$ consists of a frozen feature extractor and trainable spatial heads applied to activations received at each layer of feature extractor. In Fig. 11 we show a schematic view of the spatial heads. Each spatial head receives as input the activation from layer $i$ of feature extractor along with guidance scale ($s_T^{emb}$), image guidance scale ($s_I^{emb}$) and diffusion timestep ($t^{emb}$) embeddings. We condition the output of each spatial head on the prompt embedding $c_T^{emb}$.

**Training.** Here we provide a detailed description of our training procedure for each stage in our pipeline. Throughout the finetuning stages we use InstructPix2PixBrooks et al. (2022) dataset, batch size of 512 at $256 \times 256$ resolution and Adam Optimizer Kingma & Ba (2017).

**Stage 3.** Our multimodal guidance distillation algorithm is specified in Alg. 1 of the main manuscript. For training, we use a learning rate of $10^{-5}$. We train on a single A100 GPU and training takes around 14 hours.

**Stage 4.** Image-editing UNet model is an $\epsilon - prediction$ model. However, we find that prior to adversarial training, finetuning the UNet model for $v - prediction$ helped to reduce artifacts during training. We use a teacher-student set-up and train the student to match the $v - prediction$ from multimodal guidance distilled teacher obtained in Stage 3 (We obtain $v - prediction$ from $\epsilon - prediction$ using the standard equation as in Salimans & Ho (2022a)). For training, we use a learning rate of $10^{-6}$ with 4k warm-up steps. Training takes about a day.

Our adversarial distillation pipeline is shown in Fig. 3 and the corresponding training algorithm is specified in Alg. 2 of the main manuscript. We train for 20k iterations and learning rate of $10^{-5}$. Training takes about 3 days on a single A100 GPU.

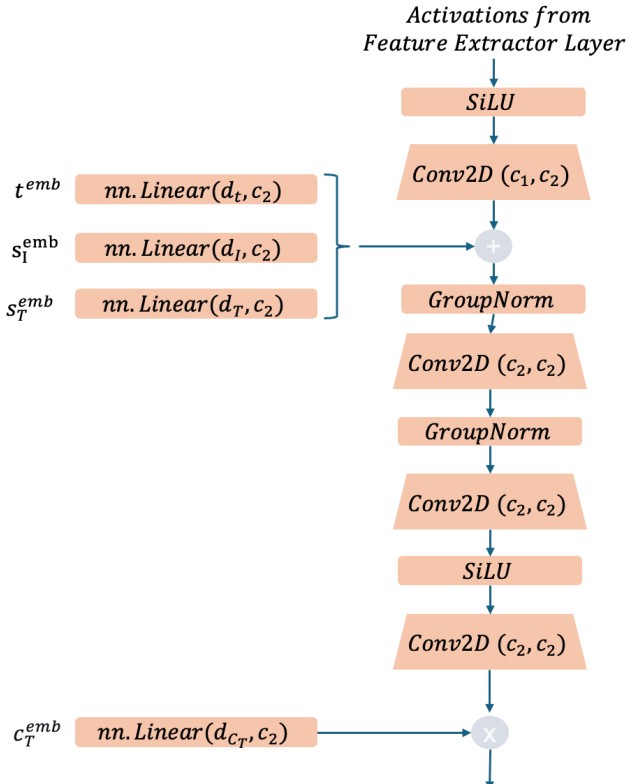

Figure 11: Adversarial Head Architecture.

Here we add further details regarding our training pipeline. We provide details of each training component here.

- **Teacher.** Teacher model is initialized from the UNet checkpoint obtained in **Stage 3** and remains frozen. We evaluate the teacher model for 5 sampling steps to obtain the clean latent $x_0$ using $LCM$ scheduler Luo et al. (2023). We find that the teacher checkpoint generates good quality results for 5 sampling steps.

- **Student.** Student model is initialized from the UNet checkpoint obtained after the $v-$ $prediction$ finetuning described above. Student is trained to denoise the latent and generate a clean latent $\hat{x}_0$ at each sampled diffusion timestep $t$. During training, we use $Euler\,Discrete$ Karras et al. (2022b) noise scheduler with 8 training timesteps. During evaluation, we evaluate the student for a single timestep using $LCM$ scheduler.

- **Discriminator.** Discriminator consists of Feature Extractor and spatial heads. We only train the spatial heads while the feature extractor remains frozen. Discriminator uses a separate $Euler\,Discrete$ Karras et al. (2022b) noise scheduler with 1000 training timesteps. During training, we find that fixing discriminator noise distribution to logit-normal as in Zhang et al. (2024b) and using $mean = -1, std = 1$ gave us the best performance. We ablate this property in Section 4.3.

## A.3 ADDITIONAL TEMPORAL CONSISTENCY EVALUATION

MoViE enforces temporal consistency through cross-frame attention by concatenating keys and values from one or more anchor frames. Similar to StreamV2V Liang et al. (2025), we use CLIPFrame and Warp Error Lai et al. (2018) to additionally evaluate temporal consistency on DAVIS validation dataset (see Tab. 3). We find that MoViE achieves competitive temporal consistency while being significantly more efficient compared to models with more sophisticated temporal mechanisms, such as TokenFlow Geyer et al. (2023) and Rerender-A-Video Yang et al. (2023), which rely on inversion, dense feature matching, and warping across frames.

| Method | CLIPFrame↑ | Warp Error↓ | TFLOPs |
|---|---|---|---|
| TokenFlow Geyer et al. (2023) | 97.04 | 114.25 | 109.35 |
| Rerender-A-Video Yang et al. (2023) | 96.20 | 107.00 | 107.52 |
| StreamV2V Liang et al. (2025) | 96.58 | 102.99 | 2.60 |
| MoViE [Ours] | 96.82 | 107.96 | 0.76 |

Table 3: Additional temporal consistency and TFLOPs reported on DAVIS validation dataset.

| Component | Ablation Type | Ablation Detail | CLIP-Image | TFLOPs |
|---|---|---|---|---|
| U-Net | Architecture Optimizations | None | 0.174 | 18.75 |
| | | Token Merging Li et al. (2024a) | 0.74 | 17.01 |
| | | Feature Caching Ma et al. (2023) | 0.54 | 16.24 |
| | | Cross Attention Pruning Li et al. (2024b) | 0.73 | 16.01 |
| | Model Quantization | + DFQ Nagel et al. (2019) | 0.6 | 4.00 |
| | | + Adaround Nagel et al. (2020) | 0.6 | 4.00 |

Table 4: Ablation study on U-Net architecture optimization.

## A.4 ABLATIONS DETAILS

**U-Net Architecture Optimizations.**     For U-Net optimization, we ablate design choices in Tab. 4 by optimizing U-Net with popular techniques such as token merging Li et al. (2024a), feature caching Ma et al. (2023), and cross-attention pruning Li et al. (2024b). We find that pruning cross-attention layers at the highest resolution as proposed by Li et al. (2024b) offers a more favorable quality vs. TFLOPs trade-off compared to the alternatives. Additionally, we also ablate quantization approaches for optimization of U-Net. We evaluated 8-bit quantization on top of the cross-attention pruned U-Net using two post-training quantization techniques, namely DFQ Nagel et al. (2019) and Adaround Nagel et al. (2020). It resulted in a degraded performance, likely due to the high dynamic range of attention weights, which may require higher bit widths for accurate representation.

