# OpenReview forum: "MoViE: Mobile Diffusion for Video Editing"
_ICLR.cc/2026/Conference — ICLR 2026 Conference Withdrawn Submission_

### Official Review · Reviewer_kL7D · 2025-10-29

**Soundness:** 2
**Presentation:** 3
**Contribution:** 2
**Rating:** 4
**Confidence:** 3

**Summary:**

This paper presents MoViE, a diffusion-based video editing framework optimized for mobile devices, achieving real-time (12 FPS) performance through architectural and algorithmic distillation. It introduces Mobile-Pix2Pix, multimodal guidance distillation, and adversarial step distillation to reduce computation by over 10× while preserving controllability. Overall, the work makes an impressive contribution toward bringing efficient diffusion video editing to edge devices.

**Strengths:**

**Significant efficiency gains**: The proposed methods deliver remarkable acceleration (up to 12 FPS on mobile), achieving over 10× speed-up compared to prior diffusion-based video editing approaches while maintaining controllability.
The work meaningfully advances the feasibility of deploying diffusion-based video editing models on edge devices.

**Clear and well-presented writing**: The paper is clearly written, logically structured, and easy to follow. Figures and tables are informative and effectively illustrate both the qualitative and quantitative improvements.

**Weaknesses:**

**Sacrificed editing quality and limited trade-off analysis between efficiency and quality**:
While the acceleration results are impressive, the editing quality is noticeably sacrificed. As shown in Fig. 6 and Fig. 9, although the outputs generally align with the target prompts, the background consistency and preservation of non-edited attributes are suboptimal. The paper would benefit from a clearer analysis of how editing quality degrades with increasing speed. A figure similar to Fig. 5, explicitly visualizing the trade-off between efficiency and edit quality across different stages or methods, would substantially strengthen the discussion.

**Insufficient quantitative evaluation of edit quality**: The quantitative experiments rely mainly on CLIP-based metrics, which only partially capture editability, reconstruction fidelity, and video quality. Including additional task-specific or perceptual metrics—such as FVD, LPIPS, or user preference scores focused on temporal consistency—would make the evaluation more comprehensive.

**Questions:**

**Temporal consistency and flickering artifacts**:
Since the base model (InstructPix2Pix) is originally designed for image editing, applying it to video editing may introduce frame-to-frame inconsistencies such as flickering or temporal jitter. How does the proposed method handle or regularize temporal consistency during inference, especially after single-step adversarial distillation?

---

### Official Review · Reviewer_btFq · 2025-10-29

**Soundness:** 3
**Presentation:** 2
**Contribution:** 2
**Rating:** 4
**Confidence:** 5

**Summary:**

This paper introduces MoViE, a diffusion-based video editing framework optimized for on-device real-time performance. The authors identify major computational bottlenecks in existing diffusion pipelines—namely, the large UNet architecture, iterative sampling, and multi-pass classifier-free guidance—and propose a series of orthogonal optimizations to overcome them. The first component, Mobile-Pix2Pix, reduces architectural overhead by pruning high-resolution attention layers and replacing the standard variational autoencoder with the Tiny Autoencoder for Stable Diffusion (TAESD). The second, Multimodal Guidance Distillation, merges text and image guidance into a single forward pass, cutting the number of forward evaluations per diffusion step by threefold. Finally, Adversarial Step Distillation compresses multi-step denoising into a single step while preserving controllability by conditioning the student model on teacher latents and corresponding guidance scales. Collectively, these innovations enable 12 FPS video editing on mobile devices, achieving a 10×–20× reduction in latency with minimal degradation in output quality.

**Strengths:**

Mobile-Pix2Pix achieves substantial efficiency gains without noticeable perceptual degradation by pruning high-resolution attention layers and replacing the standard VAE with the Tiny Autoencoder for Stable Diffusion (TAESD), a lightweight deterministic model trained using adversarial and reconstruction objectives.

Multimodal Guidance Distillation extends classifier-free guidance to jointly handle text and image modalities. The guidance scales are embedded into the UNet’s ResNet blocks, allowing the model to perform a single forward pass per diffusion step while preserving controllable guidance strength during inference.

The Adversarial Step Distillation framework compresses multi-step denoising into a single step while maintaining guidance controllability. Unlike prior adversarial distillation approaches such as LADD, which lose control flexibility, this method conditions both the student and discriminator on the teacher’s latents and the corresponding guidance scales.

Finally, the evaluation is comprehensive and convincing, incorporating CLIP-based metrics, PickScore, CLIPFrame, and Warp Error to assess temporal consistency. A human evaluation further confirms that users prefer MoViE’s results over baselines such as TokenFlow, InsV2V, and Rerender-A-Video. Overall, the paper demonstrates technical rigor, practical relevance, and a significant advancement in efficient and controllable video editing for mobile platforms.

**Weaknesses:**

I view the multimodal guidance distillation as a direct extension of Meng et al. [1]. The paper would benefit from a deeper discussion of any non-trivial technical challenges or unique insights encountered when generalizing this approach from a single text modality to both text and image modalities. Such clarification would help distinguish the contribution beyond an incremental adaptation.

Competing models such as TokenFlow are evaluated at 50 diffusion steps. Considering that this paper’s main focus is efficiency, it would be valuable to include results for 20-step (or fewer) baselines to present a more balanced view of the latency–performance trade-off.

Table 1 would also be more informative if it reported GPU or memory usage. On-device efficiency depends not only on latency but also on the memory footprint, which is currently unreported and is crucial for assessing real-world deployability.

Please see the questions part.

**Questions:**

I was one of the reviewers of the ICCV version of this paper and noticed that Table 1 appears to be identical to the previous submission, except for the new clarification that “best-suited prompts” were used for competing methods. Could the authors clarify whether the same prompt settings were already used in the previous version, or if the results have changed under the updated evaluation setup? It would also be helpful if the authors could provide concrete examples of what “best-suited prompts” mean in practice—for instance, examples of how a descriptive prompt or an instructional prompt differs across methods such as TokenFlow, InsV2V, or Rerender-A-Video. This clarification would make the fairness and consistency of the qualitative and quantitative comparisons much clearer.

---

### Official Review · Reviewer_EZTm · 2025-11-01

**Soundness:** 3
**Presentation:** 3
**Contribution:** 3
**Rating:** 4
**Confidence:** 4

**Summary:**

This paper proposes MoViE, an on-device text-guided video editing pipeline that targets ~12 fps on recent smartphones. The proposed approach is a staged optimization of a pix2pix-style editing pipeline: (1) UNet surgery (drop high-res attentions) and a lightweight autoencoder (TAESD) to cut FLOPs; (2) multimodal classifier-free guidance (CFG) distillation so the UNet consumes text/image guidance scales in a single forward; (3) adversarial step distillation to a single-step student while preserving guidance controllability (edit strength). This work achieves 3× speedup from guidance distill, ~10× from step distill, and 12 fps on Xiaomi 14 Pro / Galaxy S25 Ultra, with modest quality loss on TGVE.

**Strengths:**

1. This work proposes a comprehensive and end-to-end workflow that enables on-device video editing, which is well motivated and an important research direction.

2. The proposed Multimodal CFG distillation extends single-modality distillation to text+image with explicit scale inputs, and is simple to adopt in other editing pipelines.

3. The paper is clearly written and easy to follow. The related work section is comprehensive, and the method is elaborated clearly in detailed workflows.

**Weaknesses:**

1. This work is more engineering-oriented. Most investigated components are well-established, including efficient VAE, CFG distillation, and adversarial step distillation.

2. One big concern is the actual editing quality. The edited videos (e.g., color/style change, adding objects, altering weather, etc.) look semantically correct in the sense that the edit direction roughly follows the prompt (e.g., “make it snow,” “turn day to night”). But the texture detail, lighting continuity, and temporal coherence across frames are visibly coarse. The textures appear smoothed and low-frequency, and motion boundaries often flicker or “slide” slightly, especially on human and object contours. Compared to modern diffusion-based video editors (e.g., TokenFlow, Rerender-A-Video, VideoComposer, or AnimateDiff-based pipelines), the outputs here look simpler, flatter, and GAN-like—closer to style-transfer outputs circa 2018–2020 than to high-fidelity diffusion videos. To this end, it is a bit hard to tell whether trading this much quality for speed is worth it, i.e., maybe GAN Unets can handle it?

**Questions:**

Can you provide an end-to-end latency breakdown, including VAE decoding and other overheads?

---

### Note · Authors · 2025-11-28

I have read and agree with the venue's withdrawal policy on behalf of myself and my co-authors.